# Learning to Search in Branch-and-Bound Algorithms*

**He He    Hal Daumé III**
Department of Computer Science
University of Maryland
College Park, MD 20740
{hhe,hal}@cs.umd.edu

**Jason Eisner**
Department of Computer Science
Johns Hopkins University
Baltimore, MD 21218
jason@cs.jhu.edu

## Abstract

Branch-and-bound is a widely used method in combinatorial optimization, including mixed integer programming, structured prediction and MAP inference. While most work has been focused on developing problem-specific techniques, little is known about how to systematically design the node searching strategy on a branch-and-bound tree. We address the key challenge of learning an *adaptive* node searching order for any class of problem solvable by branch-and-bound. Our strategies are learned by imitation learning. We apply our algorithm to linear programming based branch-and-bound for solving mixed integer programs (MIP). We compare our method with one of the fastest open-source solvers, SCIP; and a very efficient commercial solver, Gurobi. We demonstrate that our approach achieves better solutions faster on four MIP libraries.

## 1   Introduction

Branch-and-bound (B&B) [1] is a systematic enumerative method for global optimization of non-convex and combinatorial problems. In the machine learning community, B&B has been used as an inference tool in MAP estimation [2, 3]. In applied domains, it has been applied to the "inference" stage of structured prediction problems (e.g., dependency parsing [4, 5], scene understanding [6], ancestral sequence reconstruction [7]). B&B recursively divides the feasible set of a problem into disjoint subsets, organized in a tree structure, where each node represents a subproblem that searches only the subset at that node. If computing bounds on a subproblem does not rule out the possibility that its subset contains the optimal solution, the subset can be further partitioned ("branched") as needed. A crucial question in B&B is how to specify the order in which nodes are considered. An effective node ordering strategy guides the search to promising areas in the tree and improves the chance of quickly finding a good incumbent solution, which can be used to rule out other nodes. Unfortunately, no theoretically guaranteed general solution for node ordering is currently known.

Instead of designing node ordering heuristics manually for each problem type, we propose to speed up B&B search by automatically learning search heuristics that are *adapted* to a family of problems.

- **Non-problem-dependent learning.** While our approach learns problem-specific policies, it can be applied to any family of problems solvable by the B&B framework. We use imitation learning to automatically learn the heuristics, free of the trial-and-error tuning and rule design by domain experts in most B&B algorithms.

- **Dynamic decision-making.** Our decision-making process is adaptive on three scales. First, it learns different strategies for different problem types. Second, within a problem type, it can evaluate the hardness of a problem instance based on features describing the solving progress. Third, within a problem instance, it adapts the searching strategy to different levels of the B&B tree and makes decisions based on node-specific features.

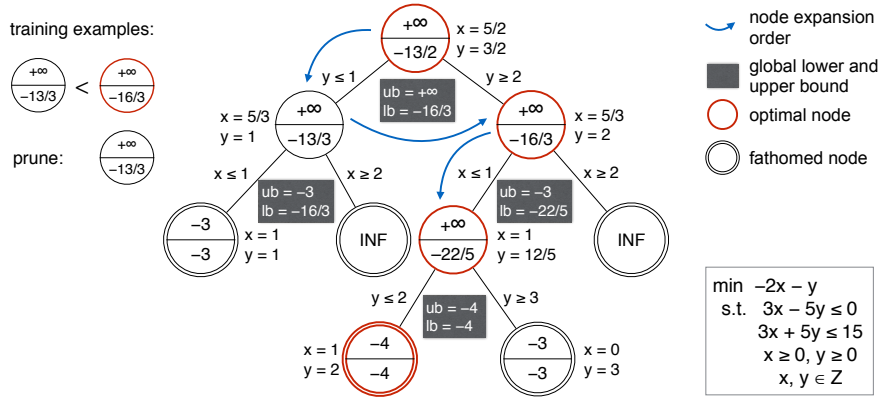

Figure 1: Using branch-and-bound to solve an integer linear programming minimization.

- **Easy incorporation of heuristics.** Most hand-designed strategies handle only a few heuristics, and they set weights on different heuristics by domain knowledge or manual experimentation. In our model, multiple heuristics can be simply plugged in as state features for the policy, allowing a hybrid "heuristic" to be learned effectively.

We assume that a small set of *solved* problems are given at training time and the problems to be solved at test time are of the same type. We learn a node selection policy and a node pruning policy from solving the training problems. The node selection policy repeatedly picks a node from the queue of all unexplored nodes, and the node pruning policy decides if the popped node is worth expanding. We formulate B&B search as a sequential decision-making process. We design a simple oracle that knows the optimal solution in advance and only expands nodes containing the optimal solution. We then use imitation learning to learn policies that mimic the oracle's behavior without perfect information; these policies must even mimic how the oracle would act in states that the oracle would not itself reach, as such states may be encountered at test time. We apply our approach to linear programming (LP) based B&B for solving mixed integer linear programming (MILP) problems, and achieve better solutions faster on 4 MILP problem libraries than Gurobi, a recent fast commercial solver competitive with Cplex, and SCIP, one of the fastest open-source solvers [8].

## 2  The Branch-and-Bound Framework: An Application in Mixed Integer Linear Programming

Consider an optimization problem of minimizing $f$ over a feasible set $\mathcal{F}$, where $\mathcal{F}$ is usually discrete. B&B uses a divide  and conquer strategy: $\mathcal{F}$ is recursively divided into its subsets $\mathcal{F}_1, \mathcal{F}_2, \ldots, \mathcal{F}_p$ such that $\mathcal{F} = \bigcup_{i=1}^{p} \mathcal{F}_i$. The recursion tree is an enumeration tree of all feasible solutions, whose nodes are subproblems and edges are the partition conditions. Slightly abusing notation, we will use $\mathcal{F}_i$ to refer to both the subset and its corresponding B&B node from now on. A (convex) relaxation of each subproblem is solved to provide an upper/lower bound for that node and its descendants. We denote the upper and lower bound at node $i$ by $\ell_{ub}(\mathcal{F}_i)$ and $\ell_{lb}(\mathcal{F}_i)$ respectively where $\ell_{ub}$ and $\ell_{lb}$ are bounding functions.

A common setting where B&B is ubiquitously applied is MILP. A MILP optimization problem has linear objective and constraints, and also requires specified variables to be integer. We assume we are minimizing the objective function in MILP from now on. At each node, we drop the integrality constraints and solve its LP relaxation. We present a concrete example in Figure 1. The optimization problem is shown in the lower right corner. At node $i$, a local lower bound (shown in lower half of each circle) is found by the LP solver. A local upper bound (shown in upper part of the circle) is available if a feasible solution is found at this node. We automatically get an upper bound if the LP solution happens to be integer feasible, or we may obtain it by heuristics.

B&B maintains a queue $\mathcal{L}$ of *active* nodes, starting with a single root node on it. At each step, we pop a node $\mathcal{F}_i$ from $\mathcal{L}$ using a node selection strategy, and compute its bounds. A node $\mathcal{F}_i$

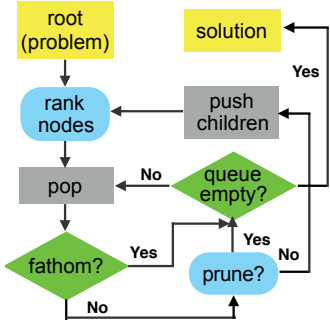

**Algorithm 1** Policy Learning $(\pi_S^*, \pi_P^*)$

$\pi_P^{(1)} = \pi_P^*, \pi_S^{(1)} = \pi_S^*, \mathcal{D}_S = \{\}, \mathcal{D}_P = \{\}$
**for** $k = 1$ **to** $N$ **do**
    **for** $Q$ **in** problem set $\mathcal{Q}$ **do**
        $\mathcal{D}_S^{(Q)}, \mathcal{D}_P^{(Q)} \leftarrow \text{COLLECTEXAMPLE}(Q, \pi_P^{(k)}, \pi_S^{(k)})$
        $\mathcal{D}_S \leftarrow \mathcal{D}_S \cup \mathcal{D}_S^{(Q)}, \quad \mathcal{D}_P \leftarrow \mathcal{D}_P \cup \mathcal{D}_P^{(Q)}$
        $\pi_S^{(k+1)}, \pi_P^{(k+1)} \leftarrow$ train classifiers using $\mathcal{D}_S$ and $\mathcal{D}_P$
    **return** Best $\pi_S^{(k)}, \pi_P^{(k)}$ on dev set

Figure 2: **Our method at runtime (left) and the policy learning algorithm (right).** *Left*: our policy-guided branch-and-bound search. Procedures in the rounded rectangles (shown in blue) are executed by policies. *Right*: the DAgger learning algorithm. We start by using oracle policies $\pi_S^*$ and $\pi_P^*$ to solve problems in $\mathcal{Q}$ and collect examples along oracle trajectories. In each iteration, we retrain our policies on all examples collected so far (training sets $\mathcal{D}_D$ and $\mathcal{D}_S$), then collect additional examples by running the newly learned policies. The COLLECTEXAMPLE procedure is described in Algorithm 2.

is *fathomed* (i.e., no further exploration in its subtree) if one of the following cases is true: (a) $\ell_{lb}(\mathcal{F}_i)$ is larger than the current global upper bound, which means all solutions in its subtree can not possibly be better than the incumbent; (b) $\ell_{lb}(\mathcal{F}_i) = \ell_{ub}(\mathcal{F}_i)$; at this point, B&B has found the best solution in the current subtree; (c) The subproblem is infeasible. In Figure 1, fathomed nodes are shown in double circles and infeasible nodes are labeled by "INF".

If a node is not fathomed, it is branched into children of $\mathcal{F}_i$ that are pushed onto $\mathcal{L}$. Branching conditions are shown next to each edge in Figure 1. The algorithm terminates when $\mathcal{L}$ is empty or the gap between the global upper bound and lower bound achieves a specified tolerance level. In the example in Figure 1, we follow a DFS order. Starting from the root node, the blue arrows points to the next node popped from $\mathcal{L}$ to be branched. Updated global lower and upper bounds after a node expansion is shown on the board under each branched node.

## 3 Learning Control Policies for Branch-and-Bound

A good search strategy should find a good incumbent solution early and identify non-promising nodes before they are expanded. However, naively applying a single heuristic through the whole process ignores the dynamic structure of the B&B tree. For example, DFS should only be used at nodes that promise to lead to a good feasible solution that may replace the incumbent. Best-bound-first search can quickly discard unpromising nodes, but should not be used frequently at the top levels of the tree since the bound estimate is not accurate enough yet. Therefore, we propose to learn policies adaptive to different problem types and different solving stages.

There are two goals in a B&B search: finding the optimal solution and proving its optimality. There is a trade-off between the two goals: we may be able to return the optimal solution faster if we do not invest the time to prove that all other solutions are worse. Thus, we will aim only to search for a "good" (possibly optimal) solution without a rigorous proof of optimality. This allows us to prune unpromising portions of the search tree more aggressively. In addition, obtaining a certificate of optimality is usually of secondary priority for practical purposes.

We assume the branching strategy and the bounding functions are given. We guide search on the enumeration tree by two policies. Recall that B&B maintains a priority queue of all nodes to be expanded. The *node selection policy* determines the priorities used. Once the highest-priority node is popped, the *node pruning policy* decides whether to discard or expand it given the current progress of the solver. This process continues iteratively until the tree is empty or the gap reaches some specified tolerance. All other techniques used during usual branch-and-bound search can still be applied with our method. The process is shown in Figure 3.

**Oracle.** Imitation learning requires an oracle at training time to demonstrate the desired behavior. Our ideal oracle would expand nodes in an order that minimized the number of node expansions subject to finding the optimal solution. In real branch-and-bound systems, however, the optimal sequence of expanded nodes cannot be obtained without substantial computation. After all, the effect of expanding one node depends not only on local information such as the local bounds it obtains, but also on how many pruned nodes it may lead to and many other interacting strategies such as branching variable selection. Therefore, given our single goal of finding a good solution quickly, we design an oracle that finds the optimal solution without a proof of optimality. We assume optimal solutions are given for training problems.[1] Our node selection oracle $\pi_S^*$ will always expand the node whose feasible set contains the optimal solution. We call such a node an *optimal node*. For example, in Figure 1, the oracle knows beforehand that the optimal solution is $x = 1, y = 2$, thus it will only search along edges $y \geq 2$ and $x \leq 1$; the optimal nodes are shown in red circles. All other non-optimal nodes are fathomed by the node pruning oracle $\pi_P^*$, if not already fathomed by standard rules discussed in Section 2. We denote the optimal node at depth $d$ by $\mathcal{F}_d^*$ where $d \in [0, D]$ and $\mathcal{F}_0^*$ is the root node.

**Imitation Learning.** We formulate the above approach as a sequential decision-making process, defined by a state space $\mathcal{S}$, an action space $\mathcal{A}$ and a policy space $\Pi$. A trajectory consists of a sequence of states $s_1, s_2, \ldots, s_T$ and actions $a_1, a_2, \ldots, a_T$. A policy $\pi \in \Pi$ maps a state to an action: $\pi(s_t) = a_t$. In our B&B setting, $\mathcal{S}$ is the whole tree of nodes visited so far, with the bounds computed at these nodes. The node selection policy $\pi_S$ has an action space {*select node* $\mathcal{F}_i$: $\mathcal{F}_i \in$ *queue of active nodes*}, which depends on the current state $s_t$. The node pruning policy $\pi_P$ is a binary classifier that predicts a class in {*prune*, *expand*}, given $s_t$ and the most recently selected node (the policy is only applied when this node was not fathomed). At training time, the oracle provides an optimal action $a^*$ for any possible state $s \in \mathcal{S}$. Our goal is to learn a policy that mimics the oracle's actions along the trajectory of states encountered by the policy. Let $\phi \colon \mathcal{F}_i \to \mathbb{R}^p$ and $\psi \colon \mathcal{F}_i \to \mathbb{R}^q$ be feature maps for $\pi_S$ and $\pi_P$ respectively. The imitation problem can be reduced to supervised learning [9, 10, 11]: the policy (classifier/regressor) takes a feature-vector description of the state $s_t$ and attempts to predict the oracle action $a_t^*$.

A generic node selection policy assigns a score to each active node and pops the highest-scoring one. For example, DFS uses a node's depth as its score; best-bound-first search uses a node's lower bound as its score. Following this scheme, we define the score of a node $i$ as $\mathbf{w}^T \phi(\mathcal{F}_i)$ and $\pi_S(s_t) = $ *select node* $\arg\max_{\mathcal{F}_i \in \mathcal{L}} \mathbf{w}^T \phi(\mathcal{F}_i)$, where $\mathbf{w}$ is a learned weight vector and $\mathcal{L}$ is the queue of active nodes. We obtain $\mathbf{w}$ by learning a linear ranking function that defines a total order on the set of nodes on the priority queue: $\mathbf{w}^T \left(\phi(\mathcal{F}_i) - \phi(\mathcal{F}_{i'})\right) > 0$ if $\mathcal{F}_i > \mathcal{F}_{i'}$. During training, we only specify the order between optimal nodes and non-optimal nodes. However, at test time, a total order is obtained by the classifier's automatic generalization: non-optimal nodes close to optimal nodes in the feature space will be ranked higher.

DAgger is an iterative imitation learning algorithm. It repeatedly retrains the policy to make decisions that agree better with the oracle's decisions, in those situations that were encountered when running past versions of the policy. Thus, it learns to deal well with a realistic distribution of situations that may actually arise at test time. Our training algorithm is shown in Algorithm 1. Algorithm 2 illustrates how we collect examples during B&B. In words, when pushing an optimal node to the queue, we want it ranked higher than all nodes currently on the queue; when pushing a non-optimal node, we want it ranked lower than the optimal node on the queue if there is one (note that at any time there can be at most one optimal node on the queue); when popping a node from the queue, we want it pruned if it is not optimal. In the left part of Figure 1, we show training examples collected from the oracle policy.

## 4 Analysis

We show that our method has the following upper bound on the expected number of branches.

**Theorem 1.** *Given a node selection policy which ranks some non-optimal node higher than an optimal node with probability $\epsilon$, a node pruning policy which expands a non-optimal node with probability $\epsilon_1$ and prunes an optimal node with probablity $\epsilon_2$, assuming $\epsilon, \epsilon_1, \epsilon_2 \in [0, 0.5]$ under the*

**Algorithm 2** Running B&B policies and collect example for problem $Q$

---

**procedure** COLLECTEXAMPLE($Q, \pi_S, \pi_P$)
$\quad \mathcal{L} = \{\mathcal{F}_0^{(Q)}\}$, training set $\mathcal{D}_S^{(Q)} = \{\}, \mathcal{D}_P^{(Q)} = \{\}, i \leftarrow 0$
$\quad$ **while** $\mathcal{L} \neq \emptyset$ **do**
$\quad\quad \mathcal{F}_k^{(Q)} \leftarrow \pi_S$ pops a node from $\mathcal{L}$,
$\quad\quad$ **if** $\mathcal{F}_k^{(Q)}$ is optimal **then** $\mathcal{D}_P^{(Q)} \leftarrow \mathcal{D}_P^{(Q)} \cup \left\{ \left(\psi(\mathcal{F}_k^{(Q)}), expand\right)\right\}$
$\quad\quad$ **else** $\mathcal{D}_P^{(Q)} \leftarrow \mathcal{D}_P^{(Q)} \cup \left\{ \left(\psi(\mathcal{F}_k^{(Q)}), prune\right)\right\}$
$\quad\quad$ **if** $\mathcal{F}_k^{(Q)}$ is not fathomed **and** $\pi_P(\mathcal{F}_k^{(Q)}) = expand$ **then**
$\quad\quad\quad \mathcal{F}_{i+1}^{(Q)}, \mathcal{F}_{i+2}^{(Q)} \leftarrow$ expand $\mathcal{F}_k^{(Q)}, \quad \mathcal{L} \leftarrow \mathcal{L} \cup \{\mathcal{F}_{i+1}^{(Q)}, \mathcal{F}_{i+2}^{(Q)}\}, i \leftarrow i + 2$
$\quad\quad\quad$ **if** an optimal node $\mathcal{F}_d^{*(A)} \in \mathcal{L}$ **then**
$\quad\quad\quad\quad \mathcal{D}_S^{(Q)} \leftarrow \mathcal{D}_S^{(Q)} \cup \left\{ \left(\phi(\mathcal{F}_d^{*(Q)}) - \phi(\mathcal{F}_{i'}^{(Q)}), 1\right) : \mathcal{F}_{i'}^{(Q)} \in \mathcal{L} \text{ and } \mathcal{F}_{i'}^{(Q)} \neq \mathcal{F}_d^{(Q)*}\right\}$
$\quad$ **return** $\mathcal{D}_S^{(Q)}, \mathcal{D}_P^{(Q)}$

---

*policy's state distribution, we have*

$$\text{expected number of branches} \leq \left( \sigma(\epsilon, \epsilon_1, \epsilon_2) \sum_{d=0}^{D} (1-\epsilon_2)^d + (1-\epsilon_2)^{D+1} \frac{(1-\epsilon)\epsilon_1}{1-2\epsilon_1} + 1\right) D,$$

*where* $\sigma(\epsilon, \epsilon_1, \epsilon_2) = \left(\frac{1-\epsilon_2}{1-2\epsilon\epsilon_1} + \frac{\epsilon_2}{1-2\epsilon_1}\right)\epsilon\epsilon_1$.

Let the optimal node at depth $d$ be $\mathcal{F}_d^*$. Note that at each push step, there is at most one optimal node on the queue. Consider a queue having one optimal node $\mathcal{F}_d^*$ and $m$ non-optimal nodes ranked before the optimal one. The following lemma is useful in our proof:

**Lemma 1.** *The average number of pops before we get to* $\mathcal{F}_d^*$ *is* $\frac{m}{1-2\epsilon\epsilon_1}$, *among which the number of branches is* $N_B(m, \text{opt}) = \frac{m\epsilon_1}{1-2\epsilon\epsilon_1}$, *and the number of non-optimal nodes pushed after* $\mathcal{F}_d^*$ *is* $N_{\text{push}}(m, \text{opt}) = \frac{m\epsilon_1}{1-2\epsilon\epsilon_1}\left[2(1-\epsilon)^2 + 2\epsilon(1-\epsilon)\right] = \frac{2m\epsilon_1(1-\epsilon)}{1-2\epsilon\epsilon_1}$, *where* opt *indicates the situation where one optimal node is on the queue.*

Consider a queue having no optimal node and $m$ non-optimal nodes, which means an optimal internal node has been pruned or the optimal leaf has been found. We have

**Lemma 2.** *The average number of pops to empty the queue is* $\frac{m}{1-2\epsilon_1}$, *among which the number of branches is* $N_B(m, \overline{\text{opt}}) = \frac{m\epsilon_1}{1-2\epsilon_1}$, *where* $\overline{\text{opt}}$ *indicates the situation where no optimal node is on the queue.*

Proofs of the above two lemmas are given in Appendix A.

Let $T(M_d, \mathcal{F}_d^*)$ denote the number of branches until the queue is empty, after pushing $\mathcal{F}_d^*$ to a queue with $M_d$ nodes. The total number of branches during the B&B process is $T(0, \mathcal{F}_0^*)$. When pushing $\mathcal{F}_d^*$, we compare it with all $M$ nodes on the queue, and the number of non-optimal nodes ranked before it follows a binomial distribution $m_d \sim \text{Bin}(\epsilon, M_d)$. We then have the following two cases: (a) $\mathcal{F}_d^*$ will be pruned with probability $\epsilon_2$: the expected number of branches is $N_B(m_d, \overline{\text{opt}})$; (b) $\mathcal{F}_d^*$ will not be pruned with probability $1 - \epsilon_2$: we first pop all nodes before $\mathcal{F}_d^*$, resulting in $N_{\text{push}}(m_d, \text{opt})$ new nodes after it; we then expand $\mathcal{F}_d^*$, get $\mathcal{F}_{d+1}^*$, and push it on a queue with $M_{d+1} = N_{\text{push}}(m_d, opt) + M_d - m_d + 1$ nodes. Thus the total expected number of branches is $N_B(m_d, \text{opt}) + T(M_{d+1}, \mathcal{F}_{d+1}^*)$.

The recursion equation is

$$T(M_d, \mathcal{F}_d^*) = \mathbb{E}_{m_d \sim \text{Bin}(\epsilon, M_d)}\left[(1-\epsilon_2)\left(N_B(m_d, \text{opt}) + 1 + T(M_{d+1}, \mathcal{F}_{d+1}^*)\right) + \epsilon_2 N_B(M_d, \overline{\text{opt}})\right].$$

At termination, we have

$$T(M_D, \mathcal{F}_D^*) = \mathbb{E}_{m_D \sim \text{Bin}(\epsilon, M_D)}\left[(1-\epsilon_2)\left(N_B(m_D, \text{opt}) + N_B(M_D - m_D, \overline{\text{opt}})\right) + \epsilon_2 N_B(M_D, \overline{\text{opt}})\right].$$

Note that we ignore node fathoming in this recursion. The path of optimal nodes may stop at $\mathcal{F}_d^*$ where $d<D$, thus $T(M_d, \mathcal{F}_d^*)$ is an upper bound of the actual expected number of branches. The expectation over $m_d$ can be computed by replacing $m_d$ by $\epsilon M_d$ since all terms are linear in $m_d$. Solving for $T(0, \mathcal{F}_0^*)$ gives the upper bound in Theorem 1. Details are given in Appendix B.

For the oracle, $\epsilon=\epsilon_1=\epsilon_2=0$ and it branches at most $D$ times when solving a problem. For non-optimal policies, as for all pruning-based methods, our method bears the risk of missing the optimal solution. The depth at which the first optimal node is pruned follows a geometric distribution and its mean is $1/\epsilon_2$. In practice, we can put higher weight on the class *prune* to learn a high-precision classifier (smaller $\epsilon_2$).

## 5   Experiments

**Datasets.** We apply our method to LP-based B&B for solving MILP problems. We use four problem libraries suggested in [12]. MIK[2] [13] is a set of MILP problems with knapsack constraints. Regions and Hybrid are sets of problems of determining the winner of a combinatorial auction, generated from different distributions by the Combinatorial Auction Test Suite (CATS)[3] [14]. CORLAT [15] is a real dataset used for the construction of a wildlife corridor for grizzly bears in the Northern Rockies region. The number of variables ranges from 300 to over 1000; the number of constraints ranges from 100 to 500. Each problem set is split into training, test and development sets. Details of the datasets are presented in Appendix C. For each problem, we run SCIP until optimality, and take the (single) returned solution to be the optimal one for purposes of training. We exclude problems which are solved at the root in our experiment.

**Policy learning.** For each problem set, we split its training set into equal-sized subsets randomly and run DAgger on one subset in each iteration until we have taken two passes over the entire set. Too many passes may result in overfitting for policies in later iterations. We use LIBLINEAR [16] in the step of training classifiers in Algorithm 1. Since mistakes during early stages of the search are more serious, our training places higher weight on examples from nodes closer to the root for both policies. More specifically, the example weights at each level of the B&B tree decay exponentially at rate $2.68/D$ where $D$ is the maximum depth[4], corresponding to the fact that the subtree size increases exponentially. For pruning policy training, we put a higher weight (tuned from $\{1, 2, 4, 8\}$) on the class *prune* to counter data imbalance and to learn a high-precision classifier as discussed earlier. The class weight and SVM's penalty parameter $C$ are tuned for each library on its development set.

The features we used can be categorized into three groups: (a) node features, computed from the current node, including lower bound[5], estimated objective, depth, whether it is a child/sibling of the last processed node; (b) branching features, computed from the branching variable leading to the current node, including pseudocost, difference between the variable's value in the current LP solution and the root LP solution, difference between its value and its current bound; (c) tree features, computed from the B&B tree, including global upper and lower bounds, integrality gap, number of solutions found, whether the gap is infinite. The node selection policy includes primarily node features and branching feature, and the node pruning policy includes primarily branching features and tree features. To combine these features with depth of the node, we partition the tree into 10 uniform levels, and features at each level are stacked together. Since the range of objective values varies largely across problems, we normalize features related to the bound by dividing its actual value by the root node's LP objective. All of the above features are cheap to obtain. Actually they use information recorded by most solvers , thus do not result in much overhead.

**Results.** We compare with SCIP (Version 3.1.0) (using Cplex 12.6 as the LP solver), and Gurobi (Version 5.6.2). SCIP's default node selection strategy switches between depth-first search and best-first search according a plunging depth computed online. Gurobi applies different strategies (including pruning) for subtrees rooted at different nodes [17, 18]. Both solvers adopt the branch-

| Dataset | Ours | | | Ours (prune only) | | | SCIP (time) | | Gurobi (node) | |
|---|---|---|---|---|---|---|---|---|---|---|
| | speed | OGap | IGap | speed | OGap | IGap | OGap | IGap | OGap | IGap |
| MIK | **4.69×** | **0.04‰** | **2.29%** | 4.45× | **0.04‰** | **2.29%** | 3.02‰ | **1.89%** | 0.45‰ | 2.99% |
| Regions | 2.30× | 7.21‰ | **3.52%** | 2.45× | 7.68‰ | 3.58% | **6.80‰** | **3.48%** | 21.94‰ | 5.67% |
| Hybrid | **1.15×** | **0.00‰** | **3.22%** | 1.02× | **0.00‰** | 3.55% | 0.79‰ | 4.76% | 3.97‰ | 5.20% |
| CORLAT | 1.63× | **8.99%** | 22.64% | **4.44×** | **8.91%** | **17.62%** | 6.67% | fail | 2.67% | fail |

Table 1: **Performance on solving MILP problems from four libraries.** We compare two versions of our algorithm (one with both search and pruning policies and one with only the pruning policy) with SCIP with a node limit (SCIP (node)) and Gurobi with a time limit (Gurobi (time)). We report results on three measures: speedup with respect to SCIP in default setting, the optimality gap (OGap), computed as the percentage difference between the best objective value found and the optimal objective value, the integrality gap (IGap), computed as the percentage difference between the upper and lower bounds. Here "fail" means the solver cannot find a feasible solution. The numbers are averaged over all instances in each dataset. Bolded scores are statistically tied with the best score according to a $t$-test with rejection threshold 0.05.

and-cut framework combined with presolvers and primal heuristics. Our solver is implemented based on SCIP and also calls Cplex 12.6 to solve LPs.

We compare runtime with SCIP in its default setting, which does not terminate before a proved status (e.g. solved, infeasible, unbounded). To compare the tradeoff between runtime and solution quality, we first run our dynamic B&B algorithm and obtain the average runtime; we then run SCIP with the same time limit. Since runtime is rather implementation-dependent and Gurobi is about four times faster than SCIP [8], we use the number of nodes explored as time measure for Gurobi. As Gurobi and SCIP apply roughly the same techniques (e.g. cutting-plane generation, heuristics) at each node, we believe fewer nodes explored implies runtime improvement had we implemented our algorithm based on Gurobi. Similarly, we set Gurobi's node limit to the average number of nodes explored by our algorithm.

The results are summarized in Table 1. Our method speeds up SCIP up to a factor of 4.7 with less than 1% loss in objectives of the found solutions on most datasets. On CORLAT, the loss is larger (within 10%) since these problems are generally harder; both SCIP and Gurobi failed to find even one feasible solution given a time/node limit on some problems. Note that SCIP in its default setting works better on Regions and Hybrid, and Gurobi better on the other two, while our adaptive solver performs well consistently. This shows that effectiveness of strategies are indeed problem dependent.

**Ablation analysis.** To assess the effect of node selection and pruning separately, we report details of their classification performance in Tabel 2. Both policies cost negligible time compared with the total runtime. We also show result of our method with the pruning policy *only* in Table 1. We can see that the major contribution comes from pruning. We believe there are two main reasons: a) there may not be enough information in the features to differentiate an optimal node from non-optimal ones; b) the effect of node selection may be covered by other interacting techniques, for instance, a non-optimal node could lead to better bounds due to the application of cutting planes.

**Informative features.** We rank features on each level of the tree according to the absolute values of their weights for each library. Although different problem sets have its own specific weights and rankings of features, a general pattern is that closer to the top of the tree the node selection policy prefers nodes which are children of the most recently solved node (resembles DFS) and have better bounds; in lower levels it still prefers deeper nodes but also relies on pseudocosts of the branching variable and estimates of the node's objective, since these features get more accurate as the search goes deeper. The node pruning policy tends to not pruning when there are few solutions found and the gap is infinite; it also relies much on differences between the branching variable's value, its value in the root LP solution and its current bound.

**Cross generalization.** To testify that our method learns strategies specific to the problem type, we apply the learned policies across datasets, i.e., using policies trained on dataset A to solve problems in dataset B. We plot the result as a heatmap in Figure 3, using a measure combining runtime and the

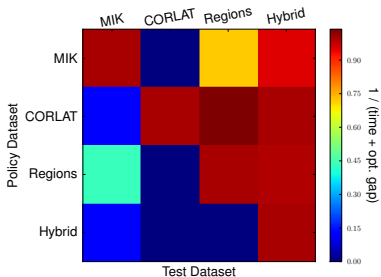

Figure 3: **Performance of policies cross datasets.** The y-axis shows datasets on which a policy is trained. The x-axis shows datasets on which a policy is tested. Each block shows $1/(\text{runtime}+\text{optimality gap})$, where runtime and gap are scaled to $[0,1]$ for experiments on the same test dataset. Values in each row are normalized by the diagonal element on that row.

Table 2: **Classification performance of the node selection and pruning policy.** We report the percentage of nodes pruned (prune rate), false positive (FP) and false negative (FN) error rate of the pruning policy, comparison error of the selection policy (only for comparisons between one optimal and one non-optimal node), as well as the percentage of time used on decision making.

| Dataset | prune rate | prune err | | comp err | time (%) | |
|---|---|---|---|---|---|---|
| | | FP | FN | | select | prune |
| MIK | 0.48 | 0.01 | 0.46 | 0.34 | 0.02 | 0.04 |
| Regions | 0.55 | 0.20 | 0.32 | 0.32 | 0.00 | 0.00 |
| Hybrid | 0.02 | 0.00 | 0.98 | 0.44 | 0.02 | 0.02 |
| CORLAT | 0.24 | 0.00 | 0.76 | 0.80 | 0.01 | 0.01 |

optimality gap. We invert the values so that hotter blocks in the figure indicate better performance. Note that there is a hot diagonal. In addition, MIK and CORLAT are relatively unique: policies trained on other datasets lose badly there. On the other hand, Hybrid is more friendly to other policies. This probably suggests that for this library most strategies works almost equally well.

# 6  Related Work

There is a large amount of work on applying machine learning to make dynamic decisions inside a long-running solver. The idea of learning heuristic functions for combinatorial search algorithms dates back to [19, 20, 21]. Recently, [22] aims to balance load in parallel B&B by predicting the subtree size at each node. Nodes of the largest predicted subtree size are further split into smaller problems and sent to the distributed environment with other nodes in a batch. In [23], a SVM classifier is used to decide if probing (a bound tightening technique) should be used at a node in B&B. However, both prior methods handle a relatively simple setting where the model only predicts information about the current state, so that they can simply train by standard supervised learning. This is manifestly not the case for us. Since actions have influence over future states, standard supervised learning does not work as well as DAgger, an imitation learning technique that focuses on situations most likely to be encountered at test time.

Our work is also closely related to speedup learning [24], where the learner observes a solver solving problems and learns patterns from past experience to speed up future computation. [25] and [26] learned ranking functions to control beam search (a setting similar to ours) in planning and structured prediction respectively. [27] used supervised learning to imitate strong branching in B&B for solving MIP. The primary distinction in our work is that we explicitly formulate the problem as a sequential decision-making process, thus take aciton's effects on future into account. We also add the pruning step besides prioritization for further speedup.

# 7  Conclusion

We have presented a novel approach to learn an adaptive node searching order for different classes of problems in branch-and-bound algorithms. Our dynamic solver learns when to leave an unpromising area and when to stop for a good enough solution. We have demonstrated on multiple datasets that compared to a commercial solver, our approach finds solutions with a better objective and establishes a smaller gap, using less time. In the future, we intend to include a time budget in our model so that we can achieve a user-specified trade-off between solution quality and searching time. We are also interested in applying multi-task learning to transfer policies between different datasets.

## Footnotes

*This material is based upon work supported by the National Science Foundation under Grant No. 0964681.

[1]For prediction tasks, the optimal solutions usually come for free in the training set; otherwise, an off-the-shelf solver can be used.

[2]Downloaded from http://ieor.berkeley.edu/~atamturk/data

[3]Available at http://www.cs.ubc.ca/~kevinlb/CATS/

[4]The rate is chosen such that examples at depth 1 are weighted by 5 and examples at $0.6D$ by 1.

[5]If the node is a child of the most recent processed node, its LP is not solved yet and its bounds will be the same as its parent's.

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
