[Reviews · NeurIPS 2014]

Submitted by Assigned_Reviewer_13

The paper present a heuristic for node selection in Branch and Bound for Mixed Integer Programs based on machine learning.
Although machine learning used for node selection is not new the paper present a new approach (to the best of my knowledge).
They utilize a classifier together with an oracle for training two aspects: a node selection policy and a node pruning policy.
The first one is used to enforce a linear order/priority on the current open nodes of the Branch and Bound while the second one is used to further shrink the list of open nodes by pruning
the unpromising ones. The results look very promising compared to a state-of-the-art solver (Gurobi).

The paper is well written and easy to follow, however I would have liked to read more details on their experimental setup. In particular it is not clear which kind of LP solver is used together with DAgger (is it Gurobi itself or another one?). Also some tuning parameters in page 6 are reported without explaining the process to arrive to such parameters.
The related work section should also be strengthened. Here, I report a couple of references that might be worth considering:
- "A Supervised Machine Learning Approach to Variable Branching in Branch-And-Bound" by Alvarez, Louveaux, Wehenkel
- "Guiding Combinatorial Optimization with UCT" by Sabharwal, Samulowitz, Reddy

Despite the encouraging results, a more thorough experimental section would be also helpful and insightful.
For instance, it is not clear to me what is the main contribution that comes from the node selection independently from node pruning. A more extensive experimental result section showing these two aspects separetely would be insightful. Furthermore, despite the optimality gap compared to Gurobi looks indeed good, the method is inherently incomplete because of the pruning (as pointed out by the authors as well). How does it compare if only $\pi_s$ is used?
A comparison based on a larger data set (not in term of instances but in term of different classes of problems) would also be welcome addition.
Which were the features extracted for training that were the most relevant among the one you used?
Finally, the presence of an oracle knowing the optimal solution is necessary in your setup. This is rarely the case when solving real-life problems. One could easily think of running an off-the-shelf solver with a reasonable timeout on those problems and then use the best solution found for training. The question would then be: how robust is the solution proposed when used with sub-optimal oracles?

[Minor Comments]
- I would suggest the author to use a grey-scale friendly version of figures (that otherwise are difficult to follow when printed in grey-scale)
- Page 3, 3rd paragraph: there is a reference to a "Figure 3". Is it the correct reference?
- Page 4, Figure 2: I believe the state machine describing the online algorithm is missing the case in which there are no feasible solutions.
- Page 4, end of the page: "at any time there can be at most one optimal node on the queue". This is in general not true for a traditional Branch and Bound setting.
You may want to specify that this applies only to the specific setup.

Addendum: Thanks for answering a good portion of the questions above in the rebuttal.
Summary: The work is very interesting, relevant and with encouraging results. The experimental section is somewhat limited and still leaves open questions regarding the ultimate potential of the approach proposed.

Submitted by Assigned_Reviewer_22

This articles proposes a new heuristic approach for node selection and node pruning in Branch and Bound. The heuristic is adapted and based on feature weight learning from a data set of training examples. A rather "generic" theoretical analysis is given that provides a bound on the number of explored branches as a function of some "error probabilities" attached to any heuristic.
Finally, an experimental analysis, consisting in comparing two versions of the approach proposed by the authors to a state-of-the-art (but with degraded performances) solver is performed. Results (which are good) are given in terms of "quality" performance, not time performance.

The quality of the paper is good, the algorithms seem sound, have been implemented, and the theoretical analysis seems correct (even though I must confess I did not read the appendix in much detail).

The paper is also clearly written, even though some elements which may (maybe) hinder the applicability of the approach are left aside. These points are mainly linked to the time performance of the approach. Even though different implementations may result in different computation times for the same algorithm, I miss :
- Some rough idea of the time taken by the approach
- A simple complexity analysis. It is not at all clear to me whether the execution time is linear with the number of nodes developed (and not, let say, exponential in the branching factor of the B&B tree).
- Nothing is said also, about the time spent in the training phase : How many problems, of which size, are to be solved exactly in a data set before the weights used in the heuristic are tuned? Isn't it unfair not to consider at all training time in the method performance, since heuristic weights seem to be rather specific to a data set (as emphasized by the authors in Section 5)?

I think the paper contribution is original. The authors have made the effort to model the B&B heuristic choice problem as a sequential decision problem and as a learning problem. I would be curious to read about the applicability of feature-based reinforcement learning approaches, which are rather standard to solve such problems (and learn the weights of your IL heuristic).

The significance is very hard to assess, without any time performance information...
Summary: This paper proposes a nice "learning" approach for heuristic. It is clearly written and I enjoyed reading it. However, I fill a bit "ill at ease" not to be able to assess how the approach scales, compared to non-degraded, state-of-the art approaches to MILP... Therefore I cannot give a good "impact" score.

Submitted by Assigned_Reviewer_24

This paper provides a strategy to learn adaptive node searching orders in branch and bound algorithms. The authors motivate this through an application of B&B on mixed Integer-Linear programs.

I am not an expert in B&B algorithms, and cannot comment on the utility of this work in the light of existing algorithms. However, this paper seems theoretically sound, and seems to solve an important problem of automatically learning adaptive strategies for node searching. Most existing works focus on heuristics, many of which are often problem specific. This paper seems to apply learning techniques to an optimization strategy, which is an interesting idea.

Summary: On a whole, I think this paper contributes a novel framework of utilizing machine learning in learning adaptive policies for B&B algorithms. I am not an expert, but this paper seems theoretically sound.
Author Feedback
Author rebuttal: Reviewer 13

Ablation Analysis:
The two policies are equally important to the success of our method. If we turn off pruning and use ONLY THE SELECTION POLICY, the policy may search deep down a wrong branch before getting to the optimal node, which is what we saw in the preliminary experiments and is reflected by the theoretical analysis as well. This also greatly increases training time. Conversely, we’ve done a quick experiment on the MIK dataset where ONLY A PRUNING POLICY IS LEARNED, taking best-first search as the selection policy. The result shows that with a similar number of nodes explored, the optimality gap is about three times larger than that with both policies learned. The node ranking error rate of best-first search is also much higher than the learned selection policy (0.22 vs 0.12). We will include these results and analysis in the final version.

Sub-optimal oracle:
The learning algorithm gives a performance bound relative to the oracle, which means that the quality of the oracle is relevant but we still have guarantees even with a suboptimal oracle.
In fact, if the oracle is not too noisy, our learner may still generalize in a way that discriminates good and bad nodes better than the oracle does. As suggested, we will add experiments with a suboptimal oracle to test this.
Note also that even if our proposed method finds a suboptimal solution, a fallback strategy is to use this (fast) solution to initialize a commercial solver.

Feature selection:
Based on feature weights in the learned policy, features related to the objective value and the bound improvement are most helpful. The degree of importance also varies across different depth.(Of course, looking at raw feature weights can be misleading; in the final version we will use a more principled method to determine which features are most useful, e.g., feature selection via different strengths of L1 regularization.)

Experiment setup:
We used Gurobi for solving the LPs. The class imbalance weights are tuned on the dev set. The example weight decay parameter is set such that examples at the root have weight 5 and those at the maximum depth have weight 1. The number of policy learning iterations was chosen based on previous work using DAgger.

Related work:
We thank the reviewer for the references and will discuss in the final version. Alvarez et al. used standard supervised learning to imitate strong branching. Actually we also tried that, but with node selection and pruning, adding variable branching does not give us much gain. Sabharwal et al. combined Monte Carlo tree search into B&B. This is similar to a RL approach, where part of the test time is used for exploration; while we take the imitation learning approach: exploration is only needed during training. The UCT score uses the LP objective to estimate a node’s quality, while our approach uses a flexible feature representation which can incorporate more heuristics.

Reviewer 22

Running time:
On average our implementation is an order of magnitude slower than Gurobi (on the MIK dataset it's 22s vs. 0.8s). However, the decision time (feature calculation and scoring) is only 0.004s, which means most of the slowness is due to a less sophisticated implementation of the B&B framework.
We will definitely include the runtime of our approach in the final draft. The only overhead in our method is computing the features and scores for each node. The features are mostly simple calculation with the LP solver's result at each node; in fact, most of them are also computed (merely for logging purposes) by the competing MILP solver (Gurobi). The scores are dot-products, hence fast to compute.

Complexity:
Just as for classical B&B solvers, the runtime is essentially linear on the number of nodes developed. (However, this is not precise since the time spent on a node involves solving the LP at that node, whose difficulty varies from node to node.)

Training time:
We were assuming a setting where the training time can be amortized over a large number of test problems from the same (stationary) distribution as the training problems. (For example, an auction server might run hundreds of combinatorial auctions every day, or a machine translation system might translate hundreds of thousands of sentences every day.)
But you are correct that not all settings are like this; so in the final version we will report performance as a function of training time. In the current experiment, roughly half the problems are used for training and half for testing. In preliminary experiments, reducing the number of training examples didn’t seem to lead to substantial degradation in performance.

Comparison to RL algorithms:
We have tried least square policy iteration in preliminary experiments, but did not get good results. We suspect that given the exponential number of possible rankings of nodes, such methods would require a significant amount of exploration (long training time) to learn decent policies.

Again, we thank all reviewers for the many good suggestions and we will address them in the final version.